# Versatile Reactivity of Mn^II^ Complexes in Reactions with N-Donor Heterocycles: Metamorphosis of Labile Homometallic Pivalates vs. Assembling of Endurable Heterometallic Acetates

**DOI:** 10.3390/molecules26041021

**Published:** 2021-02-15

**Authors:** Ruslan A. Polunin, Igor S. Evstifeev, Olivier Cador, Stéphane Golhen, Konstantin S. Gavrilenko, Anton S. Lytvynenko, Nikolay N. Efimov, Vadim V. Minin, Artem S. Bogomyakov, Lahcène Ouahab, Sergey V. Kolotilov, Mikhail A. Kiskin, Igor L. Eremenko

**Affiliations:** 1L. V. Pisarzhevskii Institute of Physical Chemistry of the National Academy of Sciences of Ukraine, Prospekt Nauki 31, 03028 Kiev, Ukraine; 1111111@gmail.com; anton.s.lytvynenko@gmail.com; 2N. S. Kurnakov Institute of General and Inorganic Chemistry, Russian Academy of Sciences, Leninsky Prosp. 31, 119991 Moscow, GSP-1, Russia; i.evstifeev@gmail.com (I.S.E.); nnefimov@yandex.ru (N.N.E.); minin@igic.ras.ru (V.V.M.); ilerem@igic.ras.ru (I.L.E.); 3University of Rennes, CNRS, Institut des Sciences Chimiques de Rennes (ISCR)–UMR 6226, F-35000 Rennes, France; stephane.golhen@univ-rennes1.fr (S.G.); lahcene.ouahab@univ-rennes1.fr (L.O.); 4Research-And-Education ChemBioCenter, National Taras Shevchenko University of Kyiv, Chervonotkatska str., 61, 03022 Kiev, Ukraine; kgavrio@gmail.com; 5Enamine Ltd. 78 Chervonotkatska str., 02660 Kiev, Ukraine; 6International Tomography Center, Siberia Branch of Russian Academy of Science, Institutskaya str. 3a, 630090 Novosibirsk, Russia; bus@tomo.nsc.ru

**Keywords:** polynuclear complexes, manganese, pyridines, coordination polymers, porous materials, magnetic properties

## Abstract

Reaction of 2,2′-bipyridine (2,2′-bipy) or 1,10-phenantroline (phen) with [Mn(Piv)_2_(EtOH)]*_n_* led to the formation of binuclear complexes [Mn_2_(Piv)_4_L_2_] (L = 2,2′-bipy (**1**), phen (**2**); Piv^−^ is the anion of pivalic acid). Oxidation of **1** or **2** by air oxygen resulted in the formation of tetranuclear Mn^II/III^ complexes [Mn_4_O_2_(Piv)_6_L_2_] (L = 2,2′-bipy (**3**), phen (**4**)). The hexanuclear complex [Mn_6_(OH)_2_(Piv)_10_(pym)_4_] (**5**) was formed in the reaction of [Mn(Piv)_2_(EtOH)]*_n_* with pyrimidine (pym), while oxidation of **5** produced the coordination polymer [Mn_6_O_2_(Piv)_10_(pym)_2_]*_n_* (**6**). Use of pyrazine (pz) instead of pyrimidine led to the 2D-coordination polymer [Mn_4_(OH)(Piv)_7_(µ_2_-pz)_2_]*_n_* (**7**). Interaction of [Mn(Piv)_2_(EtOH)]*_n_* with FeCl_3_ resulted in the formation of the hexanuclear complex [Mn^II^_4_Fe^III^_2_O_2_(Piv)_10_(MeCN)_2_(HPiv)_2_] (**8**). The reactions of [MnFe_2_O(OAc)_6_(H_2_O)_3_] with 4,4′-bipyridine (4,4′-bipy) or *trans*-1,2-(4-pyridyl)ethylene (bpe) led to the formation of 1D-polymers [MnFe_2_O(OAc)_6_L_2_]*_n_*·2*n*DMF, where L = 4,4′-bipy (**9**·2DMF), bpe (**10**·2DMF) and [MnFe_2_O(OAc)_6_(bpe)(DMF)]*_n_*·3.5*n*DMF (**11**·3.5DMF). All complexes were characterized by single-crystal X-ray diffraction. Desolvation of **11**·3.5DMF led to a collapse of the porous crystal lattice that was confirmed by PXRD and N_2_ sorption measurements, while alcohol adsorption led to porous structure restoration. Weak antiferromagnetic exchange was found in the case of binuclear Mn^II^ complexes (*J_Mn-Mn_* = −1.03 cm^−1^ for **1** and **2**). According to magnetic data analysis (*J_Mn-Mn_* = −(2.69 ÷ 0.42) cm^−1^) and DFT calculations (*J_Mn-Mn_* = −(6.9 ÷ 0.9) cm^−1^) weak antiferromagnetic coupling between Mn^II^ ions also occurred in the tetranuclear {Mn_4_(OH)(Piv)_7_} unit of the 2D polymer **7**. In contrast, strong antiferromagnetic coupling was found in oxo-bridged trinuclear fragment {MnFe_2_O(OAc)_6_} in **11**·3.5DMF (*J_Fe-Fe_* = −57.8 cm^−1^, *J_Fe-Mn_* = −20.12 cm^−1^).

## 1. Introduction

Polynuclear coordination compounds of transition metals are widely used as catalysts in various reactions [1,2,3,4,5], as starting materials for the preparation of nanosized oxides or metals [6,7,8], as well as the basis for the creation of new magnetic materials [9,10,11,12,13] and coordination polymers with remarkable properties [14,15,16,17,18,19,20]. In the majority of these applications the stability or reactivity of the polynuclear core play an important role. In many cases polynuclear complexes undergo rearrangement or dissociation in solution because of instability in solvent or upon interaction with the “additional” ligands [21,22,23,24,25,26,27], and the resulting compounds may contain a metal core different from the one existing in the starting complex. Such rearrangement reactions lead to the formation of certain polynuclear cores, which are most stable under the reaction conditions. For example, binuclear cores M_2_(O_2_CR)_4_ (M^II^ = Cu, Zn, Co, Ni) [28], trinuclear cores Fe^III^_3_ or Fe^III^_2_M^II^ (M = Ni, Co, Mn) [29,30,31,32], M^II^_2_Ln^III^ (M^II^ = Zn, Co, Ni; Ln is lanthanide) [33,34,35] often form due to their exceptional stability in various media. However, the result of such rearrangements is not always predictable, since many factors may influence the reaction pathway.

In this work manganese complexes were chosen as the objects of research due to some specific features of this ion, which are manifested in the reactivity of its polynuclear complexes [25,26]. The reasons for such differences include the wide range of stable oxidation states of manganese with low energy barriers for redox-transformations [36,37], as well as, in the case of Mn^II^, the relatively high kinetic lability of this ion [36]. The additional reasons for interest to homo- and heterometallic manganese complexes is that such species have found applications as building blocks for the synthesis of single-molecule magnets [9,15,38] or various coordination polymers capable of absorbing guest molecules [39,40]. It is also known that molecular manganese compounds can oligomerize upon crystallization during solvent changes, giving rise to complexes with higher nuclearity [9,26,26].

The aim of this study was to reveal the stability limits of Mn^II^ homometallic (pivalate) or heterometallic (Fe_2_Mn acetate) complexes in reactions with N-donor ligands (Scheme 1) and to compare the reactivity of Mn^II^ species with the reactivity of Ni^II^ and Co^II^ analogues. An additional aim of the study was to see the influence of the structure of new compounds on their ability to uptake guest molecules or on the magnetic exchange interactions in the polymetallic cores.

In this study we used pivalate manganese(II) polymer [Mn(Piv)_2_(EtOH)]*_n_* and trinuclear manganese(II)-iron(III) oxoacetate [MnFe_2_O(OAc)_6_(H_2_O)_3_]*_n_* (Piv^−^ is the anion of pivalic acid) as starting compounds. Reactions of [Mn(Piv)_2_(EtOH)]*_n_* with both chelating (2,2′-bipyridine (2,2′-bipy) and 1,10-phenantroline (phen)) and non-chelating bridging (pyrimidine (pym), pyrazine (pz)) N-donor ligands in the absence/presence of oxygen as oxidant were investigated. In contrast to Ni^II^ and Co^II^, the reaction of Mn^II^ pivalate with FeCl_3_ led to formation of a compound with a Mn_4_Fe_2_O_2_(Piv)_10_ core. Reaction of the same Mn^II^ pivalate with a N-donor ligand led to the assembly of various polynuclear cores, while reaction of Fe^III^_2_Mn^II^ acetate with N-donors of the pyridine type led to the formation of new complexes where the trinuclear Fe^III^_2_Mn^II^ core was preserved. The crystal and molecular structures of eleven new compounds were determined—[Mn_2_(Piv)_4_L_2_] (L = 2,2′-bipy (**1**), phen (**2**)), [Mn_4_O_2_(Piv)_6_L_2_]·MeCN (L = 2,2′-bipy (**3**·MeCN), phen (**4**·0.5MeCN)), [Mn_6_(OH)_2_(Piv)_10_(pym)_4_] (**5**), [Mn_6_O_2_(Piv)_10_(pym)_2_]*_n_* (**6**), [Mn_4_(OH)(Piv)_7_(pz)_2_]*_n_*∙2*n*MeCN (**7**∙2MeCN), [Mn_4_Fe_2_O_2_(Piv)_10_(MeCN)_2_(HPiv)_2_]·2MeCN (**8**·2MeCN), [MnFe_2_O(OAc)_6_L_2_]_n_·2*n*DMF (L = 4,4′-bipy (**9**·2DMF), bpe (**10**·2DMF), [MnFe_2_O(OAc)_6_(bpe)(DMF)]*_n_*·4*n*DMF (**11**·3.5DMF). Sorption of alcohols by porous coordination polymers, as well as the magnetic properties of several polynuclear Mn-containing complexes were studied.

## 2. Results and Discussion

### 2.1. Synthesis

The reaction of [Mn(Piv)_2_(EtOH)]*_n_* and 2,2′-bipyridine (2,2′-bipy) or 1,10-phenantroline (phen) in MeCN under an argon atmosphere led to formation of molecular complexes [Mn_2_^II^(Piv)_4_L_2_] (L = 2,2′-bipy (**1**), phen (**2**), Figure 1). Oxidation of these complexes or the initial reaction system resulted in rearrangement of the dinuclear molecules to tetranuclear complexes [Mn_2_^II^Mn_2_^III^O_2_(Piv)_6_L_2_] (L = 2,2′-bipy (**3**), phen (**4**)).

Binuclear complexes **1** and **2** are similar to reported binuclear complexes of other *d*-metals with +2 charge [41,42,43,44,45,46,47,48]. The complex **3** is known and was obtained earlier by reaction of [Mn(Piv)_2_(EtOH)]*_n_* and 2,2′-bipy in THF in air [49].

Reaction of [Mn(Piv)_2_(EtOH)]*_n_* and pyrimidine (pym) in MeCN under an argon atmosphere led to the molecular hexanuclear complex [Mn_6_(OH)_2_(Piv)_10_(pym)_4_] (**5**). On exposure to air in MeCN compound **5** was oxidized to give the 1D polymer [Mn^II^_4_Mn^III^_2_(µ_4_-O)_2_(Piv)_10_(µ_2_-pym)(pym)]*_n_* (**6**).

Previously reported carboxylate complexes of transition metals ions formed polymeric compounds with pyrimidine [50], so the formation of the molecular complex **5** is uncommon. At the same time, the formation of the mixed valence hexanuclear fragment {Mn^II^_4_Mn^III^_2_(µ_4_-O)_2_(Piv)_10_} is similar to that in **5** and quite typical, for example, oxidation of [Mn(Piv)_2_(EtOH)]*_n_* (the same starting compound as in formation of **5**) by air led to [Mn^II^_4_Mn^III^_2_(µ_4_-O)_2_(Piv)_10_(HPiv)(EtOH)_3_] [51]; a similar hexanuclear Mn_6_ core was found in polymeric compounds [40,52]. Introduction of pyrimidine ligands resulted in formation of new coordination polymers.

The use of pyrazine (pz) instead of pyrimidine in the reaction with [Mn(Piv)_2_(EtOH)]*_n_* in MeCN under an argon atmosphere led to the formation of the 2D-coordination polymer [Mn_4_(OH)(Piv)_7_(µ_2_-pz)_2_]*_n_* (**7**). This complex was stable in air and was not soluble in MeCN, and this reason probable precluding oxidation of Mn^II^. Pyrazine is a quite typical bridging ligand in the chemistry of manganese, forming polymers with Mn^II^ carboxylates [53,54], as well as with the mixed-valence hexanuclear fragment {Mn^II^_4_Mn^III^_2_(µ_4_-O)_2_(O_2_CR)_10_} [54,55,56]. From an analysis of literature, as well as from the results of this study it can be noted that the tetranuclear unit {Mn^II^_2_Mn^III^_2_O_2_(O_2_CR)_6_} usually forms in the presence of chelating ligands, such as 2,2′-bipy and phen, while the hexanuclear unit {Mn^II^_4_Mn^III^_2_O_2_(O_2_CR)_10_} is produced in reactions with monodentate N-donor ligands or in the absence of such ligands.

It is known that the M^II^ ions (such as Mn^II^, Co^II^ or Ni^II^) and Fe^III^ ions quite typically form trinuclear acetates [Fe_2_MO(OAc)_6_(H_2_O)_3_] [30]. Earlier we reported that Co^II^ and Ni^II^ pivalates with FeCl_3_·6H_2_O in acetonitrile gave similar trinuclear complexes [Fe_2_MO(Piv)_6_(HPiv)_3_] (M = Co, Ni) [19]. Unexpectedly, [Mn(Piv)_2_(EtOH)]*_n_* reacted with FeCl_3_·6H_2_O in MeCN under an argon atmosphere with formation of the hexanuclear complex [Mn^II^_4_Fe^III^_2_O_2_(Piv)_10_(MeCN)_3_] (**8**). On the other hand, with an excess of pivalic acid in the reaction of FeCl_3_·6H_2_O, Mn(NO_3_)_2_·6H_2_O and KOH [Mn^II^Fe^III^_2_O(Piv)_6_(HPiv)_3_] is formed [57]. The trinuclear fragments {Fe_2_MnO(Piv)_6_} were also generated in situ in the synthesis of coordination polymers [58].

The synthesis of compounds **9**–**11** was based on substitution of coordinated water molecules in [Fe_2_MnO(OAc)_6_(H_2_O)_3_] by 4,4′-bipyridine (4,4′-bipy) or 1,2-*trans*-(4-pyridyl)ethene (bpe). In compounds **9** and **10** all vacancies in the coordination spheres of the metal ions are filled by the nitrogen atoms of 4,4′-bipy or bpe ligands, while 4,4′-bipy or bpe molecules act both as bridging and non-bridging (capping) ligands, as will be described in details in the X-ray structures description (vide infra). In compound **11** all bpe molecules link trinuclear blocks but only two of three possible “vacant” positions in the coordination spheres of metal ions are occupied by a pyridine group of bpe; the third position is filled by DMF.

Formation of coordination polymers **9**–**11** can be formally described as the generation of a 1D-chain of [MnFe_2_O(OAc)_6_(L)]*_n_* (L = 4,4′-bipy or bpe) and filling of the third positions by terminal 4,4′-bipy and [MnFe_2_O(OAc)_6_(4,4′-bipy)_3_]*_n_* (for **9**), by bpe or DMF (for **10** and **11**, respectively). Synthesis of compounds [M_3_O(RCO_2_)_6_(4,4′-bipy)_3_]^0/+^ which can potentially bind metal ions was reported earlier [59]. In contrast to the reaction of [Fe_2_MO(OAc)_6_(H_2_O)_3_] (M = Co, Ni) with 4,4′-bipy which led to destruction of the trinuclear blocks or to the formation of porous [MFe_2_O(OAc)_6_(4,4′-bipy)_1.5_]*_n_* coordination polymers [27], compound [Fe_2_MO(OAc)_6_(H_2_O)_3_] was stable under the same conditions and formed coordination polymers with a ratio of trinuclear block to bridging ligand equal to 1:1 or 1:2.

### 2.2. Crystal and Molecular Structures

The crystal structures of molecular complexes **1**–**5**, **8** and coordination polymers **6**, **7**, **9**–**11** were determined by single crystal X-ray analysis.

#### 2.2.1. Compounds **1** and **2**

Complexes **1** and **2** have dinuclear cores (Mn…Mn 4.448(2) and 4.015(3) Å, respectively) with a common metal-carboxylate fragment {Mn_2_(µ-Piv)_2_(η-Piv)_2_}. Despite their similar composition, the structures of these fragments are different (Figure 2a,b). The binuclear complex in **1** is centrosymmetric (the inversion center lies between the metal atoms), while compound **2** possesses axial symmetry (a 2_1_ axis passes between the metal atoms). In both **1** and **2** oxygen donors occupy four positions in the coordination sphere of each of Mn^II^ ion, and two N atoms from 2,2′-bipy or phen complete the coordination spheres of these ions to form distorted octahedra (Figure 2a,b). The Mn–O and Mn–N bond lengths in **1** and **2** fall in range 2.039(4)–2.607(6) Å and 2.263(4)–2.342(4) Å respectively, which is typical for complexes of Mn^II^ with carboxylates and 2,2′-bipy or phen [44,45,60,61,62].

To the best of our knowledge, the dinuclear core in **1** is the first example of a Mn_2_ structural unit block possessing a *syn*,*syn*-binding µ-COO-group and chelating 2,2′-bipy ligands. A compound of similar composition, Mn_2_(ad(O_2_C)_2_)_2_(2,2′-bipy)_2_·0.5H_2_O (where ad(O_2_C)_2_^2−^ is 1,3-adamantanedicarboxylate) [60] contained a dinuclear Mn_2_(µ_2_-O_2_C)_2_(η-O_2_C)_2_ unit with *syn*,*anti-*coordination mode of the µ-O_2_C-groups.

The aromatic rings of 2,2′-bipy ligands of the neighboring molecules in **1** are not parallel, and the angle between the mean planes of these molecules is 7.6(3)°. The closest distance between centroids of pairs rings of different 2,2′-bipy ligands is 3.591(4) Å (the slippage is 0.707 Å). This leads to the formation of a supramolecular chain along the *c* axis (Figure 3b), probably due to π-stacking interactions.

The aromatic rings of two phen ligands in one molecule of **2** are not parallel, and the angle between the mean planes of these molecules is 9.8(2)°. The closest distance between centroids of pairs rings (N2C18-C22, C14-C19), belonging to these different phen ligands, is 3.695(4) Å (the slippage is 1.055 Å). The mean planes of phen ligands from the neighboring different molecules of **2** are parallel and the centroids of these pairs of phen rings are located in 3.730(4) Å (the slippage is 1.561 Å). Such an arrangement of aromatic phen molecules can allow for π-stacking interactions between them and as a result formation of a supramolecular pile structure along the *c* axis (Figure 3c). However, it cannot be excluded that intramolecular π-stacking between phen molecules in **2** may be the reason for the difference between structures of this compound and **1**: the 2,2′-bipy molecules in **1** are located on different sides in respect to the inversion center, located between the Mn^II^ ions.

#### 2.2.2. Compounds **3** and **4**

The molecules of compounds **3** and **4** are centrosymmetric (an inversion center lies between the central metal ions) and possess a similar tetranuclear Mn_4_O_2_(Piv)_6_ core (Figure 3a) of a “butterfly” type, which is quite typical for Mn carboxylates [63,64,65,66]. Each Mn^III^ ion in the center of the butterfly is located in a square pyramidal coordination polyhedron (MnO_5_ chromofore, τ = 0.15 [63,64,65,66]), while Mn^II^ ions on the wings of the butterfly are hexacoordinate and donor atoms in their coordination environment form highly distorted octahedra (MnO_4_N_2_ chromophores, where N atoms belong to 2,2′-bipy or phen in **3** and **4**, respectively). Mn^III^–(µ_3_–O) and Mn^II^–(µ_3_–O) bond lengths in **3** and **4** fall in range 1.843(2)–1.851(2) Å and 2.073(3)–2.099(2) Å, respectively, which is typical for Mn-O distances within trinuclear units Mn^II^_2_Mn^III^_2_O_2_ [63,64,65,66]. Mn^III^–O and Mn^II^–O (O atoms from pivalate) bond lengths fall in range 1.957(2)–2.097(2) and 2.116(2)–2.213(3) respectively for **3**, and 1.955(3)–2.105(3) Å and 2.105(3)–2.196(3) Å respectively for **4**, making these bonds longer than the corresponding bonds of Mn ions with µ_3_-O atoms. Terminal Mn^II^ atoms fill up their coordination sphere by coordination of 2,2′–bipy or phen with Mn–N bond lengths 2.271(3), 2.284(3) Å for **3**, and 2.256(4), 2.284(4) Å for **4**.

#### 2.2.3. Compound **5**

Compound **5** crystallizes in the monoclinic space group *P*2_1_/*n* as a discrete centrosymmetric hexanuclear complex (the inversion center lies between the central metal ions Mn2, Mn3, Mn2A and Mn3A). The hexanuclear core of **5** can be considered as two identical triangular fragments {Mn_3_(OH)(Piv)_3_(pym)_2_} linked by four carboxylic acid groups (two µ_2_-Piv and two µ_3_-Piv) (Figure 4a). In each trinuclear fragment Mn^II^ ions are linked by µ_3_-OH (bond lengths Mn–O are equal to 2.049(2)–2.184(2) Å), and the O atom is located on 0.66(2) Å above the Mn1Mn2Mn3 plane, which can be an additional proof that the central oxygen atom belongs to a µ_3_-hydroxo group rather than a µ_3_-oxo (Mn_3_(µ_3_-O) unit that is expected to be planar [67,68,69,70]). One µ-O_2_C bridging group links Mn1 and Mn2, and two µ_2_-O_2_C groups link Mn1 with Mn3 (bond lengths Mn–O(Piv) and lie in the 2.095(3)–2.142(3) Å) range. In addition to the oxygen atoms of carboxylate groups, the nitrogen atoms of two pyrimidine molecules are coordinated to Mn1, completing its coordination polyhedron to form a distorted octahedron (Mn1–N bond lengths are 2.293(3) and 2.322(3) Å). Mn2 is located in the distorted tetrahedral donor set O_4_ (Mn2–O(Piv) bond lengths are in range 2.067(3)–2.097(3) Å)), and Mn3 is in a distorted octahedral donor set O_6_ (Mn3–O(Piv) bond lengths are in range 2.094(3)–2.262(2) Å)).

#### 2.2.4. Compound **6**

This compound crystallizes as a 1D polymer in the space group *Pn*, in which hexanuclear units {Mn^II^_4_Mn^III^_2_O_2_(Piv)_10_} are linked by pyrimidine bridges (Figure 4b). Generally, the structure of the Mn_6_ core in **6** is similar to that of the hexanuclear units observed in [M^II^_4_M^III^_2_(O)_2_(O_2_CR)_10_(L)_4_] complexes, where L is a neutral N- or O-donor ligands [71,72,73,74] with the difference that one Mn4 atom in **6** possesses coordination number four and is located in a coordination polyhedron, close to a distorted square-pyramid (τ = 0.12) [75]. Central Mn^III^ ions (Mn1…Mn2 2.835(3) Å) possess O_6_ donor sets (Mn–O(µ_4_–O) 1.878(7)–1.914(8) Å, Mn–O 1.933(9)–2.267(9) Å), three of four terminal Mn^II^ ions are located in O_5_N donor sets (Mn–O(µ_4_–O) 2.019(8)–2.200(8) Å, Mn–O 1.970(10)–2.457(10) Å, Mn3–N1 2.255(12), Mn5–N3 2.511(16), Mn6–N4 2.471(18) Å), where N is an atom of a bridging (N3 and N4) or non-bridging (N1) pyrimidine molecule. In the crystal lattice 1D chains of **6** are arranged parallel to the *a* axis. The crystal lattice of **6** is retained upon sample storage in air, as confirmed by powder XRD (Appendix A).

#### 2.2.5. Compound **7**

This complex crystallizes in the triclinic space group *P*-1 as a solvate with two molecules of MeCN. The tetranuclear core {Mn^II^_4_(OH)(Piv)_7_} (Figure 5a) in **7** can be described as a trinuclear µ_3_-hydroxo-centered unit Mn_3_(OH)(Piv)_4_ (Mn…Mn 3.371(2)–3.735(2) Å), linked with the fourth Mn4 ion (Mn4…Mn 3.451(2), 4.157(2) Å) by four pivalate anions. Such a Mn_4_ unit is not symmetric. The Mn_3_(OH)(Piv)_4_ bond lengths of Mn–O1M fall in the 2.131(2)–2.134(2) Å range, with atom O1M located above Mn1Mn2Mn3 plane at 0.64(3) Å, *d*(Mn–O(Piv) = 2.111(3)–2.238(3) Å. The Mn1 ion is bound to two pyrazine ligands (Mn1–N = 2.346(4), 2.389(4) Å), while the Mn3 and Mn4 ions are bound to one molecule of pyrazine (Mn3–N 2.277(4) Å, Mn4–N 2.315(4) Å). Thus, the Mn1, Mn2 and Mn3 ions possess distorted octahedral donor sets O_4_N_2_, O_6_ and O_5_N, respectively, and the Mn4 ion is located in a distorted square-pyramidal coordination environment (O_4_N donor set; τ = 0.08) [75]. All pyrazine molecules in **7** are bridging. Local symmetry centers of the unit cell are located in the centers of the pyrazine rings which link Mn1-Mn1′ and Mn3-Mn3′ ions.

Each Mn_4_ unit is connected with other four Mn_4_ units by four pyrazine bridges, while each pyrazine links two tetranuclear units. Such an arrangement results in the formation of a 2D polymer. The 2D-layers are parallel to the *bc* plane (Figure 5b,c).

#### 2.2.6. Compound **8**

The complex **8** crystallizes in the monoclinic space group *C*2/*c* as a solvate with four molecules of MeCN. The molecule of **8** has axial symmetry, with axis 2 passing between the Fe1 and Fe1A atoms through the O1M, O2M, C21, C22, C26 and C27 atoms. The structure of the complex is similar to that of the known hexanuclear complexes [Mn^II^_4_Mn^III^_2_(O)_2_(O_2_CR)_10_(L)_4_], where L is a neutral N- or O-donor ligans [71,72,73] but where the central atoms are Fe^III^ instead of Mn^III^. The central Fe^III^ ions (Fe1…Fe1A 2.8824(10) Å) possess O_6_ donor sets (Fe–O(µ_4_-O) 1.953(2), 1.962(2) Å, Fe–O(Piv) 2.020(2)–2.070(2) Å), two terminal Mn^II^ ions (Mn…Fe 3.1806(8)–3.5075(8) Å, Mn…Mn 3.4831(11), 3.4841(11) Å) are located in O_5_N donor sets (Mn–O(µ_4_-O) 2.085(2) Å, Mn–O(Piv) 2.101(3)–2.492(2) Å, Mn-N 2.343(4), where N is an atom of a MeCN molecule, and two terminal Mn^II^ ions are located in O_6_ donor sets (Mn–O(µ_4_-O) 2.086(2) Å, Mn–O(Piv) 2.104(3)–2.522(2) Å, Mn–O(HPiv) 2.240(3) (Figure 6). H-bonds are formed between of a coordinated molecule of acid and the O atom of a bridging carboxylate group (O12…O4 2.603(4) Å, O12–H 0.84 Å, O4…H 1.77 Å, angle O12–H–O4 169°).

#### 2.2.7. Compounds **9–11**

Coordination polymers **9–11** are built by linking a neutral trinuclear block, {Fe_2_MnO(OAc)_6_}, with neutral pyridine-containing bridges. The structure of the {Fe_2_MnO(OAc)_6_} unit in all these complexes is almost the same (such unit in compound **9** is shown on Figure 7 as example). In this block three metal ions (two Fe^III^ and Mn^II^) are located in the corners of an irregular triangle and generally cannot be distinguished by X-ray crystallography, so the assignment of metal ions was arbitrary. These metal ions are linked by μ_3_-O atoms and six bridging acetates, so oxygen donors occupy five positions in the coordination sphere of each metal ion. The sixth positions are taken up by a donor atom (N or O) from other ligands (4,4′-bipy, bpe or DMF), so that each metal ion is located in a distorted octahedral donor set.

M---M separations within trinuclear acetates in compounds **9**–**11** fall in range 3.240(2)–3.353(2) Å. M–(μ_3_-O) bond lengths lie in the range from 1.837(6) to 2.056(7) Å, M–O(carboxylate) bonds vary between 2.011(6) to 2.155(8) Å, which is typical for μ_3_-oxocentered carboxylates [30].

In the crystal lattice of **9** trinuclear μ_3_-oxocentered units {MnFe_2_O(OAc)_6_} are connected by 4,4′-bipy molecules. 3/4 of trinuclear blocks are linked by 4,4′-bipy with formation of a 1D zig-zag chain. Two metal ions from each Fe_2_Mn unit take part in such chain formation, while the third metal ion is bound to a non-bridging 4,4′-bipy molecule or a “terminal block” {MnFe_2_O(OAc)_6_(4,4′-bipy)_3_}. Thus, there are three types of trinuclear MnFe_2_ blocks in **9** (Figure 8):(1)MnFe_2_ units in 1D chains, bound to two bridging 4,4′-bipy and one additional non-bridging 4,4′-bipy (type A);(2)MnFe_2_ units in {MnFe_2_O(OAc)_6_(4,4′-bipy)_3_} residues (type B);(3)MnFe_2_ unit in 1D chains, bound to two bridging 4,4′-bipy and one type B unit (type C).

The distance between µ_3_-O atoms of neighboring Fe_2_Mn blocks in one 1D chain in **9** is equal to 15.222(8) Å in the case of AA blocks, 15.323 Å for AC blocks and 15.49(1) Å for BC blocks, while the angles between lines connecting the µ_3_-O atoms of the neighboring trinuclear blocks, are close to 120° (from 116.83(4)° to 120.99(2)°). Torsion angles between lines connecting four µ_3_-O atoms of adjacent Fe_2_Mn blocks, are equal to 180 ° for a CAAC fragment (i.e., all µ_3_-O atoms for this fragment belong to one plane) and ±119.64(5)° for ACAA and AACA fragments. In other words, one 1D chain of **9** turns clockwise twice by 119.64(5)°, as implied by the torsion angles for ACAA or AACA, then all µ_3_-O atoms lie in one plane in the CAAC fragment and finally the chain turns twice counterclockwise by 119.64(5)° in the ACAA and AACA fragments. The main axes of chains in compound **9** are directed along the (*c-*½*a*) vector (Figure 9).

The twofold axis passes through µ_3_-O atoms of trinuclear blocks B and C. Also the local inversion centers are located between the pyridine groups of 4,4′-bipy molecules which bind two A type trinuclear blocks. The crystal lattice of **9** does not contain continuous channels (see Appendix A).

#### 2.2.8. Complexes **10** and **11**

These are built from trinuclear units {Fe_2_MnO(OAc)_6_} bound by bpe molecules (Figure 10a). Two metal ions in each trinuclear block coordinate with pyridine rings from bridging bpe, leading to 1D-chain formation. The third metal ion is bound to a nitrogen atom of a terminal (non-bridging) bpe in **10** or oxygen atom of coordinated DMF in **11** (Figure 10b).

Chains of compound **10** are parallel and directed along the *a-*½*b* vector. The non-coordinated pyridine ring of bpe in one chain and the pyridine ring of a bridging bpe ligand from the neighboring chain are almost parallel (the angle between mean planes of these rings is 4.1(5)°), and the closest distance between these rings is 3.38(2) Å (the distance between centroids of the rings is 3.634(6) Å), the slippage is 1.179 Å, which can allow for π-interactions (Figure 11a).

Chains of compound **11** are also parallel and directed along the *c* vector (Figure 11). No specific interactions between different chains are found. Due to this peculiarities of the chain packing channels of dimensions 5 × 12 Å directed along the *a* axis form in the crystal lattice (Figure 12). Estimation of solvent-accessible volume, performed by PLATON software [76], gives a value of 36% for **11**, containing DMF molecules coordinated to metal ions, or 45% for a structure, if coordinated DMF is removed assuming that such removal does not lead to crystal lattice collapse (calculated for a probe molecule with *r* = 1.4 Å). These values correspond to ca. 0.32 cm^3^ g^−1^ pore volume, occupied by solvent in **11**, assuming that the volume occupied by coordinated DMF is not included in this value, or 0.40 cm^3^ g^−1^, if the volume of coordinated DMF is included.

### 2.3. Thermal Stability and Sorption Properties of 11·3.5DMF

The thermal stability of **11**·3.5DMF was studied by thermogravimetry. Upon heating to 275 °C, compound **11**·3.5DMF lost 26% of its weight, which corresponds to the release of both non-coordinated and coordinated solvent (Figure 13). An abrupt weight loss began at 275 °C, which was completed at 400 °C and could be associated with decomposition of the compound. The total weight loss was equal to 76.6% and corresponded to the formation of Fe_2_O_3_·1/3Mn_3_O_4_ (Figure 13a). Loss of solvent and coordinated DMF led to significant lattice disorder, as it can be concluded by comparison of the powder XRD pattern of vacuum-dried product at 145 °C and the powder XRD pattern, calculated from the single-crystal structure (Figure 13b).

For sorption experiments compound **11**·3.5DMF was heated in vacuum at 153 °C during 6 h, which led to removal of non-coordinated and coordinated DMF. The desolvated sample is hereinafter referred to as **11′**.

Compound **11′** showed only surface sorption of N_2_ or H_2_ at 78 K, which is evidence of crystal lattice collapse and is consistent with the powder XRD data. In contrast, **11′** absorbed significant quantities of methanol and ethanol at 298 K (Figure 14). Such a difference between absorption of gases and alcohols can be caused by expansion of crystal lattice of **11′** upon interaction with methanol and ethanol, similarly to reported gate-opening phenomena [77] and previously reported cases of alcohol absorption by coordination polymers [18]. Both in the cases of methanol and ethanol, the sorption capacity gradually increased to ca. 0.42 cm^3^·g^−1^ (methanol) or 0.35 cm^3^·g^−1^ (ethanol), which is in good agreement with the value of solvent-accessible volume estimated from the crystallographic data (vide supra).

The plateau in the methanol absorption isotherm at *PP*_S_^−1^ ca. 0.07–0.3 (*V*_abs._ about 0.05 cm^3^ g^−1^) corresponds to a methanol to Fe_2_Mn molar ratio 1:1 and can be associated with methanol coordination to the metal ion (in a position which was occupied by coordinated DMF in **11**). It can be concluded from the presence of such a plateau that there is a noticeable difference between the energy of methanol coordination to a metal ion in **11′** and the energy of further methanol interaction with **11′**·CH_3_OH. In contrast, a similar plateau was not found in the ethanol absorption isotherm: ethanol binding by **11′** after filling of unsaturated metal sites seems to be as efficient as ethanol binding due to its coordination (which most probably does occur). Anyhow, the maximal achieved sorption capacity of **11′** corresponds to ca. 10.5 moles of methanol or ca. 5 moles of ethanol per 1 mole of Fe_2_Mn, which is significantly higher than the sorption capacity associated with coordination.

### 2.4. EPR Spectroscopy

X-band EPR experiments for polycrystalline samples **1** and **2** were performed at 293 K. The spectra of **1** and **2** show an intense singlet without hyperfine structure with *g* ≈ 2.00. In the low magnetic field lines of low intensity are observed; their origin can be explained by the exchange interactions between paramagnetic manganese ions (Figure 15).

Since antiferromagnetic interactions with *J* = −1.03 cm^−1^ were found in **1** and **2** by magnetic data analysis (see below), all possible magnetic states of dimer of two Mn^2+^ ions with spins *S*_1,2_ = 5/2, notably, S = 0, 1, 2, 3, 4, 5 (*S* = *S*_1_ + *S*_2_) were equally populated. Furthermore, since |*J*| > *hν* ≈ 0.3 cm^−1^, transitions between states with different total spin *S* can be neglected. Thus, the spin Hamiltonian for **1** and **2** is the sum of spin Hamiltonians of five dimers with different total spins. The spin Hamiltonian (1) for single ion in **1** or **2** has a rhombic symmetry:(1)$${\hat{H}}_{i}=g_{iz}\beta H_{z}S_{iz}+g_{ix}\beta H_{x}S_{ix}+g_{iy}\beta H_{y}S_{iy}+d_{i}\cdot (S_{iz}^{2}-\frac{1}{3}S_{i}(S_{i}+1))+e_{i}\cdot (S_{ix}^{2}-S_{iy}^{2}),$$
where *g_iz_*, *g_ix_*, *g_iy_*—*z*, *x*, *y*—g-tensor components of monomer *i*, where *i* = 1, 2; *S_iz_*, *S_ix_*, *S_iy_*–projections of spin operator of monomer on coordinate axes, *S_i_* = 5/2; *d_i_*, *e_i_*—component of fine interaction tensor (so-called, single-ion). Mn^2+^ ion has half-filled d^5^ shell and S-state, so g-tensor is isotropic and close to spin-only value, so, *g_iz_* = *g_ix_* = *g_iy_* = *g* = 2.0023.

Spin Hamiltonian (2) of dimer is the sum of two Hamiltonians for interactions within mononuclear fragments of molecule and the part of their interaction:(2)$$\begin{array}{l} \hat{H}={\hat{H}}_{1}+{\hat{H}}_{2}-2JS_{1}S_{2}+d_{12}\cdot (S_{1z}\cdot S_{2z}-\frac{1}{3}S_{1}\cdot S_{2})+\\ e_{12}\cdot (S_{1x}\cdot S_{2x}-S_{1y}\cdot S_{2y}) \end{array}$$where *d*_12_, *e*_12_ are the components of fine interaction tensor, caused by dipole interaction of manganese ions.

For the total spin of dimer and neglecting of transitions between multiplets with different total spin *S = S*_1_
*+ S*_2_, spin Hamiltonian (3) can be used:(3)$$\begin{array}{l} {\hat{H}}_{S}=g\beta (H_{z}S_{z}+H_{x}S_{x}+H_{y}S_{y})+D_{S}\cdot (S_{z}^{2}-\frac{1}{3}S(S+1))+E_{S}\cdot (S_{x}^{2}-S_{y}^{2})\\ -2J(S(S+1)-S_{1}(S_{1}+1)-S_{2}(S_{2}+1)) \end{array}$$
where *D* and *E* are the components of fine interaction tensor, associated with parameters *d*_1,2_, *e*_1,2_, *d*_12_ и *e*_12_ by the following formula [78]:$$D_{S}=\alpha_{S}d_{12}+\beta_{S}d_{1}\;\;\;\;\;\;\;\;\;\;\;\;E_{S}=\alpha_{S}e_{12}+\beta_{S}e_{1}$$
(4)$$\alpha_{S}=\frac{S(S+1)+4S_{1}(S_{1}+1)}{2(2S-1)(2S+3)}\;\;\;\;\;\;\;\;\;\;\;\;\beta_{S}=\frac{3S(S+1)-3-4S_{1}(S_{1}+1)}{\left(2S-1)(2S+3\right)}$$

The spin Hamiltonian (3) was diagonalized numerically. Calculations of resonance fields of spin Hamiltonian (3) required to build a theoretical spectrum were carried out by the Belford method [79], which involves finding the values of magnetic field *H*, for which two eigenvalues of spin Hamiltonian (3) matrix, corresponding to two different eigenvectors, would differ on the *hν*.

The spin Hamiltonian parameters for compounds **1** and **2** are given in Table 1. Thus, while isotropic exchange of two dimers is same, fine interaction tensor causes a noticeable difference in the EPR spectra.

### 2.5. Magnetic Properties of Complexes ***1, 2, 7***·2MeCN and ***11***·3.5DMF

Magnetic properties of the representative complexes from the prepared series—compounds **1**, **2**, **7**·2MeCN and **11** 3.5DMF—were characterized by their temperature dependence of the molar magnetic susceptibility, *χ_M_*.

#### 2.5.1. Magnetic Properties of Complexes **1** and **2**

For both compounds *χ_M_T* values (here and below *χ_M_* is the magnetic susceptibility per formula unit and *T* is the temperature in K) monotonously decrease upon lowering the temperature from 8.17 (for **1**) or 8.64 (for **2**) cm^3^·K·mole^−1^ at 300 K to 7.81 (for **1** at 60 K) or 7.74 (for **2** at 68 K) cm^3^·K·mole^−1^, after which it falls sharply to 4.01 (for **1** at 5 K) or 0.82 (for **2** at 2 K) cm^3^·K·mole^−1^. Room-temperature values of *χ_M_T* are close to the expected spin-only value (8.75 cm^3^·K·mole^−1^ for a system with two non interacting magnetic centers with *S* = 5/2).

The spin Hamiltonian for dinuclear blocks Mn_2_ in **1** and **2** is shown as Equation (5).
(5)$$\hat{H}=-2J_{Mn-Mn}{\hat{S}}_{1}{\hat{S}}_{2}+\beta Hg_{Mn}({\hat{S}}_{1}+{\hat{S}}_{2})$$
where the first summand corresponds to the superexchange interactions between Heisenberg spins localized at metal sites (*J_Mn-Mn_*), and the second summand corresponds to the isotropic interactions between local spins and the external field through Zeeman interactions [80].

It should be noted that the spin-Hamiltonian proposed for interpretation of the magnetic properties differed from the one employed for interpretation of the EPR spectra of the same compounds. There was no contradiction between these Hamiltonians, as both of them were “partial” variations of the complete spin-Hamiltonian describing the system of the unpaired electron within the species of the compounds. This complete spin-Hamiltonian had to include all the terms: (i) Zeeman interactions with the external magnetic field; (ii) exchange interactions between the ions; (iii) zero-field splitting. However, the variations of exchange interaction parameters could not notably influence the studied EPR spectra (*vide supra*), while the influence of zero-field splitting on the magnetization curves was negligible compared to the influence of the exchange interactions. Thus, introduction of the corresponding terms into the spin-Hamiltonians and efforts to extract the corresponding parameters from the data simulations would not produce any reliable values. Regarding Zeeman interactions, their principal parameters—g-factors—were consistent (within accuracy of the methods) for the EPR and magnetochemical data.

Temperature-independent paramagnetism (*tip*) term was also introduced. Intermolecular interactions were taken into account within molecular field model (z*J*’ term).

Analytical expression for the *χ_M_T* values for the Mn_2_ unit [80] is the following:(6)$$\chi_{M}T=\frac{2Ng^{2}\mu_{B}^{2}}{k}\cdot \frac{e^{x}+5e^{3x}+14e^{6x}+30e^{10x}+55e^{15x}}{1+3e^{x}+5e^{3x}+7e^{6x}+9e^{10x}+11e^{15x}},$$
where *x* = −*J*/*kT*.

The best agreement between experimental and calculated *χ_M_T* curves (Figure 16) was achieved at parameters *J_Mn-Mn_* = −1.03(2) cm^−1^, *g_Mn_* = 2.0, z*J*’ = −0.19(1) cm^−1^ (*R*^2^ = 2.44·10^−5^) for compound **1** and *J_Mn-Mn_* = −1.03(2) cm^−1^; *g_Mn_* = 2.0, *tip* = 7·10^−4^ (*R*^2^ = 9.87·10^−5^) for compound **2** (where *R*^2^ = ∑[(*χ_M_T*)_obs._ − (*χ_M_T*)_calc._]^2^/(∑(*χ_M_T*)_obs._^2^)).

Absolute values of *J_Mn-Mn_* for **1** and **2** are higher than those reported for benzoato- and phtalato-bridged dinuclear blocks Mn_2_(µ-O_2_C)_2_(η-O_2_C)_2_ [81,82], which is consistent with the higher electron-donating ability of the *tert*-butyl groups in pivalates compared to phenyl group.

#### 2.5.2. Magnetic Properties of Complex **7** ·2MeCN

For the compound **7** 2MeCN the *χ_M_T* value monotonically decreases upon lowering the temperature from 15.78 cm^3^·K·mole^−1^ at 300 K to 0.53 cm^3^·K·mole^−1^ at 3 K. The room-temperature value of *χ_M_T* is lower than the expected spin-only value (17.5 cm^3^·K·mole^−1^ for a system with four non-interacting magnetic centers with *S* = 5/2). The coupling scheme within a tetranuclear unit is presented in Figure 17. The spin Hamiltonian for tetranuclear block Mn_4_ takes the form:(7)$$\begin{array}{c} H=-2J_{1}S_{Mn1}\cdot S_{Mn3}-2J_{2}S_{Mn1}S_{Mn2}-2J_{3}S_{Mn2}\cdot S_{Mn3}-2J_{4}S_{Mn3}S_{Mn4}\\ -2J_{5}S_{Mn2}S_{Mn4}+g_{Mn}\beta (S_{Mn1}+S_{Mn2}+S_{Mn3}+S_{Mn4}+S_{Mn5})\cdot H \end{array}$$
where the first five summands correspond to the superexchange interactions between Heisenberg spins localized at metal sites (*J*_1_–*J*_5_), and the last summand corresponds to the isotropic interactions between local spins and the external field through Zeeman interactions (*g_Mn_*), respectively [80]. A temperature-independent paramagnetism (tip) term was also introduced.

Calculation of the exchange coupling parameters were performed by full-matrix diagonalization using the Mjöllnir software [16,83]. Uncertainty values of simulation parameters were estimated as described previously [33]. Briefly, digits in brackets indicated deviation of the value, which caused 10% increase of R^2^.

The best correspondence between experimental and calculated *χ_M_T* values for compound **7** was achieved for the parameters *J*_1_ = −2.69(2) cm^−1^, *J*_2_ = −2.38(2) cm^−1^, *J*_3_ = −0.8(1) cm^−1^, *J*_4_ = −0.42(2) cm^−1^, *J*_5_ = −0.8(2) cm^−1^, *g_Mn_* = 2.0023 (fixed), *zJ*’ = −0.5(1) cm^−1^, *tip* = 0.00129 (*R*^2^ = 2.3·10^−5^).

Exchange coupling parameters have larger values for magnetic interactions of Mn1 ion with Mn2 and Mn3 which agree with the structural data: the Mn1 ion has in its coordination sphere two N atoms from two pyrazine molecules which increase its electron density and amplify the antiferromagnetic interactions. Additionally, the most effective way of magnetic interactions transfer through the OH group in the trinuclear Mn1Mn2Mn3 unit which also agrees with the received data.

The Mn_4_ units can be selected in the 2D-coordination polymer [Mn_4_(μ_3_-OH)(Piv)_7_(μ-pz)_2_]*_n_*, and exchange coupling within these units can be presented by five integrals *J*_1_*–J*_5_ (Figure 17). From the experimental *χ_M_T* vs. *T* curve the following values could be calculated by full-matrix diagonalization (performed using the Mjöllnir software [16,83]) (cm^−1^): *J*_1_ = −2.7, *J*_2_ = −2.4, *J*_3_ = −0.8, *J*_4_ = −0.4, *J*_5_ = −0.8. Signs and magnitudes of these *J* were estimated using broken-symmetry DFT calculations with TPSSh functional and LANLTZ ECPs (3d ions)/def2-SVP basis set. For calculation of each *J* value all Mn^2+^ ions in the Mn_4_ core except the two ions taking part in the coupling were “substituted” by diamagnetic Zn^2+^ ions, and then calculations of the high-spin (HS) and the broken symmetry (BS) states’ energies (see Experimental part for details of their construction) were performed. The results evidence that two exchange integrals, *J*_1_ and *J*_2_, have the same order of magnitude (−6.9 and −6.4 cm^−1^, respectively); while *J*_3_, *J*_4_, *J*_5_ fall in the range from −0.9 to −1.2 cm^−1^. The exchange through pz bridge is estimated as −0.2 cm^−1^. These results correlate with *J* values, found from *χ_M_*T vs. T curve simulation (|*J*_1_|, |*J*_2_| the highest and close to each other, |*J*_3_|*–*|*J*_5_|—lower and close to each other).

It should be noted that fitting of the *χ_M_T* vs. *T* curve for **7** 2MeCN could be fitted with simpler Hamiltonian (8):(8)$$\hat{H}=-2J_{1}(S_{1}S_{2}+S_{3}S_{4})-2J_{2}\;S_{2}S_{3}$$
with *J*_1_ = −2.08 ± 0.02 cm^−1^, *J*_2_ = −3.92 ± 0.07 cm^−1^, *g* = 2 (fixed), however there are no reasons to neglect interactions between other Mn^II^ ions, since structural features of the bridges between them are similar. The *χ_M_T* vs. *T* curve calculated with these parameters is visually the same as shown on Figure 17.

#### 2.5.3. Magnetic Properties of Complex 11·3.5DMF

For compound **11**·3.5DMF, the value of *χ_M_T* at 300 K was 4.66 cm^3^·K·mol^−1^, which is significantly lower than the expected spin-only value for three non-interacting spins 5/2 (13.125 cm^3^·K·mol^−1^). On cooling, the *χ_M_T* vs. *T* curve decreased monotonically to 3.20 cm^3^·K/mol at 50 K, after which it sharply fell to 2.08 cm^3^·K·mol^−1^ at 2 K.

The coupling scheme within a trinuclear unit is represented on Figure 18. The spin Hamiltonian for trinuclear blocks Fe_2_Mn takes the form:(9)$$\begin{array}{c} H=-2J_{Fe-Fe}S_{Fe_{1}}\cdot S_{Fe_{2}}-2J_{Fe-Mn}(S_{Fe_{1}}+S_{Fe_{2}})\cdot S_{Mn}+\\ +g_{Mn}\beta S_{Mn}\cdot H+g_{Fe}\beta (S_{Fe_{1}}+S_{Fe_{2}})\cdot H \end{array}$$
where first line corresponds to the superexchange interactions between Heisenberg spins localized at metal sites (*J_Fe-Fe_* and *J_Fe-Mn_*), the second line corresponds to the isotropic interactions between local spins and the external field through Zeeman interactions (*g_Fe_* and *g_Mn_*), respectively [80]. Intermolecular interactions were taken into account within molecular field model.

The best correspondence between experimental and calculated *χ_M_T* values for the compound was achieved at parameters *J_Fe-Fe_* = −57.8(2) cm^−1^, *J_Fe-Mn_* = −20.12(7) cm^−1^, *g_Fe_* = *g_Mn_* = 2.0023 (fixed), *zJ*’ = −0.10(1) cm^−1^, *χ*_impurities_ = 3.2% (*S*_impurities_ = 5/2).

Absolute values of *J_Fe-Mn_* and *J_Fe-Mn_* for **11** 3.5DMF are higher than those reported for the trifluoroacetate complex [Fe_2_MnO(O_2_CCF_3_)_6_(H_2_O)_3_] (*J_Fe-Fe_* = −56.50(7) cm^−1^, *J_Fe-Mn_* = −16.23(4) cm^−1^ [31], which is consistent with the higher electron-donating ability of the methyl group in acetate relative to CF_3_-group. However, for {Fe_2_MnO(Piv)_6_} blocks exchange coupling parameters [58] have close values to exchange coupling parameters for **11**·3.5DMF. It can be explained by different nitrogen ligands: pyridine group in **11**·3.5DMF in comparison with hexamethylenetetramine [58] compensates lower electron donor efficiency of acetate group in comparison with pivalate group [84].

## 3. Experimental

### 3.1. Materials and Methods

Reagents and solvents were commercially available (Sigma-Aldrich, Aldrich, St. Louis, MO, USA) and were used without further purification. Manganese pivalate [Mn(Piv)_2_(EtOH)]*_n_* and trinuclear acetate [MnFe_2_O(OAc)_6_(H_2_O)_3_], used as starting compound, were prepared as previously reported [30,51]. C,H,N-analyses were performed using a 1106 instrument (Carlo Erba, Instruments, Egelsbach, Germany). IR-spectra were measured in KBr pellets on a Spectrum BX FT-IR spectrometer (Perkin Elmer, Waltham, MA, USA) in 400–4000 cm^−1^ range. The X-ray powder diffraction analysis of **6** was carried out on a G670 (HUBER, Offenburg, Germany) Guinier camera using CuK_α1_ radiation on air. The X-ray powder diffraction analysis of **11** was performed on a D8 Advance instrument (Bruker, Billerica, MA, USA) in air.

Thermogravimetric analyses (TGA) were performed in air on Q1500 instrument, (*Paulik-Paulik-Erdey*, Budapest, Hungary). The heating rate was 5 °C per minute. Sorption of methanol and ethanol by [MnFe_2_O(OAc)_6_(dpe)] was studied gravimetrically, using a tungsten microbalance at 293 K. Each point on the absorption and desorption isotherms corresponds to equilibrium conditions (no change of sample weight at certain *p*·*p_S_*^−1^, where *p_S_* is the pressure of saturated vapor of the compound at 293 K). This sample was thermally activated at 150 °C in vacuum at 10^−2^ Torr. Volume of pores was estimated from the quantity of adsorbed alcohol using its density in liquid phase at 293 K.

Magnetic measurements were performed on a MPMS-XL (for **11**), MPMS-5S (for **7**) and PPMS (for **1** and **2**) SQUID magnetometers (Quantum Design, San Diego, CA, USA) and intrinsic diamagnetic corrections were calculated using Pascal’s constants [80]. The X-band EPR spectra for **1** and **2** were measured on a Bruker Elexsys E680-X spectrometer at *T* = 293 K.

### 3.2. Synthesis

#### 3.2.1. Synthesis of [Mn_2_(Piv)_4_(2,2′-bipy)_2_] (**1**) and [Mn_2_(Piv)_4_(phen)_2_] (**2**)

The syntheses were carried out under an argon atmosphere. [Mn(Piv)_2_(EtOH)]*_n_* (0.150 g, 0.50 mmol) was dissolved in MeCN (30 mL for **1** or 25 mL for **2**), followed by the addition of 2,2′-bipyridine (0.078 g, 0.50 mmol) for **1** or 1,10-phenantroline (0.090 g, 0.50 mmol) for **2**. The resulting colorless solution was heated at 80 °C during 30 min, then concentrated to 6–8 mL for **1** or 4–6 mL for **2** and kept at 5 °C during 24 h. Crystals were isolated by decantation, washed by cold MeCN and dried under argon stream. Yield of **1**: 0.14 g (67%), yellow crystals; **2**: 0.162 g (74%), colorless crystals. Anal, calc. for **1**, C_40_H_52_Mn_2_N_4_O_8_/found, %: C 58.1/58.0, H 6.3/6.5, N 6.8/6.9. IR-spectrum of **1** (cm^−1^): 3434 m, 3058 w, 2955 s, 2923 m, 2865 w, 1597 vs, 1542 s, 1516 s, 1482 s, 1420 s, 1373 m, 1359 m, 1225 m, 1143 w, 1101 w, 894 w, 864 w, 846 m, 792 b.w, 729 s, 637 w, 601 w. Anal, calc. for **2**, C_44_H_52_Mn_2_N_4_O_8_/found, %: C 60.4/59.8, H 6.0/6.2, N 6.4/6.4. IR-spectrum of **2** (cm^−1^): 3434 s, 2956 s, 2923 m, 2866 w, 1586 v.s, 1547 s, 1482 s, 1440 s, 1419 s, 1372 m, 1358 m, 1314 w, 1226 m, 1172 w, 1155 w, 1059 w, 1015 m, 893 m, 805 w, 792 w, 767 s, 738 m, 646 w, 625 w, 603 w, 558 w, 415 m.

#### 3.2.2. Synthesis of [Mn_4_O_2_(Piv)_6_(2,2′-bipy)_2_]·MeCN (3·MeCN) (**3**) and [Mn_4_O_2_(Piv)_6_(phen)_2_]· 0.5MeCN (4·0.5MeCN) (**4**)

The syntheses were carried out under an argon atmosphere. [Mn(Piv)_2_(EtOH)]*_n_* (0.150 g, 0.50 mmol) was dissolved in 30 mL of MeCN followed by the addition of 2,2′-bipyridine (0.078 g, 0.50 mmol) for **3·MeCN** or 1,10-phenantroline (0.090 g, 0.50 mmol) for **4·0.5MeCN**. The colorless solution was heated at 80 °C during 30 min, then kept under air at room temperature during 24 h for **3·MeCN** or 10 days for **4·0.5MeCN**. The precipitate was isolated by decantation and dissolved in 30 mL of THF, filtered and then kept under air at room temperature during one week. Brown crystals were isolated by decantation, washed by cold MeCN and dried on air. Yield of **3·MeCN**: 0.051 g (35%), brown microcrystals; yield of **4·0.5MeCN**: 0.045 g (29%), dark-brown crystals. Anal, calc. for **3·MeCN**, C_50_H_70_Mn_4_N_4_O_14_/found, %: C 51.3/51.1, H 6.0/6.2, N 4.8/4.9. IR-spectrum of **3·MeCN** (cm^−1^): 3433 m, 2956 m, 2925 m, 2868 w, 1589 v.s., 1570 s, 1482 s, 1441 m, 1415 s, 1370 s, 1357 m, 1317 w, 1225 m, 1155 w, 1016 w, 890 w, 789 w, 766 m, 740 w, 640 b.m, 431 m, 410 m. Anal, calc. for **4·0.5MeCN**, C_55_H_71.5_Mn_4_N_4.5_O_14_/found, %: C 53.3/53.4, H 5.8/5.9, N 5.1/5.2. IR-spectrum of **4·0.5MeCN** (cm^−1^): 3435 w, 2956 s, 2924 m, 2868 m, 1590 s, 1560 s, 1514 s, 1482 s, 1456 m, 1411 s, 1371 s, 1358 s, 1261 w, 1226 s, 1141 w, 1101 m, 1028 w, 891 w, 864 m, 853 m, 790 m, 732 s, 656 s, 639 s, 616 s, 420 bm.

#### 3.2.3. Synthesis of [Mn_6_(OH)_2_(Piv)_10_(pym)_4_] (**5**) and Synthesis of [Mn_6_O_2_(Piv)_10_(pym)_2_]*_n_* (**6**)

Syntheses were carried out under an argon atmosphere. [Mn(Piv)_2_(EtOH)]*_n_* (0.6 g, 1.98 mmol) was dissolved in 30 mL of EtOH, followed by the addition of pyrimidine (0.11 g, 1.40 mmol). The colorless solution was heated at 80 °C during 30 min. For the isolation of **5** the solution was concentrated to 2–3 mL and cooled at −18 °C during 24 h. For the isolation of **6** the solution was kept on air at room temperature during 24 h. The crystals were isolated by decantation, washed by cold MeCN and dried under argon stream. Yield of **5**: 0.12 g (22%) colorless crystals; yield of **6**: 0.43 g (85%), colorless crystals. Anal, calc. for **5**, C_66_H_108_Mn_6_N_8_O_22_/found, %: C 46.8/46.7, H 6.4/6.2, N 6.6/6.8. IR-spectrum of **5** (cm^−1^): 3419 b.m, 2960 s, 2928 s, 1676 s, 1587 b.s, 1484 s, 1640 s, 1422 s, 1362 s, 1227 s, 1169 w, 1077 w, 1030 w, 937 w, 895 m, 790 m, 716 m, 636 m, 601s, 559 m, 540 m, 415 m. Anal, calc. for **6**, C_58_H_98_Mn_6_N_4_O_22_/found, %: C 45.4/45.5, H 6.4/6.2, N 3.7/3.8. IR-spectrum of **6** (cm^−1^): 3434 b.m, 2958 m, 2926 w, 2870 w, 1582 v.s, 1569 s, 1482 s, 1467 w, 1417 v.s, 1374 s, 1359 s, 1228 s, 1163 w, 1076 w, 1029 w, 1003 w, 892 w, 795 w, 786 w, 715 w, 632 w, 612 m, 557 w, 437 w, 404 w.

#### 3.2.4. Synthesis of [Mn_4_(OH)(Piv)_7_(pz)_2_]*_n_*∙2*n*MeCN (**7**∙2MeCN)

The synthesis was carried out under an argon atmosphere. [Mn(Piv)_2_(EtOH)]*_n_* (0.1 g, 0.33 mmol) was dissolved in 20 mL of MeCN, then a solution containing pyrazine (0.03 g, 0.33 mmol) in 5 mL MeCN was added dropwise. The colorless solution was kept at 25 °C during 48 h. Yellow precipitate with crystals was isolated by decantation, washed by cold MeCN and dried under argon stream. Yield 0.08 g (82%). Anal, calc. for C_47_H_78_Mn_4_N_6_O_15_/found, %: C 47.6/47.4, H 6.6/6.8, N 7.1/7.2. IR-spectra (cm^−1^): 3432 b.m, 2959 s, 2928 m, 2870 w, 1589 v.s, 1569 v.s, 1483 v.s, 1458 m, 1422 s, 1374 s, 1360 s, 1227 s, 1135 w, 1047 m, 893 w, 791 m, 600 m, 564 w, 450 w, 409 m.

#### 3.2.5. Synthesis of [Mn_4_Fe_2_O_2_(Piv)_10_(MeCN)_2_(HPiv)_2_]·2MeCN (**8**·2MeCN)

The synthesis was carried out under an argon atmosphere. A reaction mixture of [Mn(Piv)_2_(EtOH)]*_n_* (0.8 g, 2.64 mmol) and FeCl_3_ (0.122 g, 0.75 mmol) in 100 mL MeCN was heated (80 °C) during 30 min to complete the dissolution of reagents. The brown solution obtained was concentrated to 50 mL and kept at room temperature during 6 h. The white and brown precipitates formed were separated from the solution. The solution was concentrated to 20 mL and kept at room temperature during 24 h. Brown crystals were isolated by decantation, washed by cold MeCN and dried under argon stream. Yield 0.16 g (15%). Anal, calc. for C_64_Fe_2_H_116_Mn_4_O_26_N_2_ (without solvent molecules)/found, %: C 46.3/46.4, H 7.0/6.9, N 1.7/1.6, Fe 6.7/6.6; Mn 13.2/13.0. IR-spectra (cm^−1^): 3192 w, 2962 s, 2930 s, 2873 m, 1692 m, 1570 s, 1484 s, 1460 m, 1420 s, 1376 s, 1360 s, 1314 w, 1227 s, 1206 m, 1031 w, 938 w, 895 m, 872 w, 787 m, 764 v.w, 604 m, 574 m, 516 m, 420 s.

#### 3.2.6. Synthesis of [MnFe_2_O(OAc)_6_(4,4′-bipy)_2_]*_n_*·2*n*DMF (**9**·2DMF)

[Fe_2_MnO(OAc)_6_(H_2_O)_3_] (0.1 g, 0.169 mmol) was dissolved in 3 mL of DMF, followed by the addition of 15 mL of MeCN, then 0.1 mL of acetic acid was added, and then bipy (0.250 g, 0.423 mmol, a 25% excess) was dissolved in this solution. After one day black crystals formed, which were collected by filtration, washed with MeCN (2 portions by 3 mL) and dried on air. Yield 0.04 g (25%). Anal, calc. for C_38_H_48_N_6_O_15_Fe_2_Mn/found, %: C 45.9/45.6, H 4.86/4.50, N 8.44/8.50.

#### 3.2.7. Synthesis of [MnFe_2_O(OAc)_6_(bpe)_2_]*_n_*·2*n*DMF (10·2DMF)

[Fe_2_MnO(OAc)_6_(H_2_O)_3_] (0.1 g, 0.169 mmol) was dissolved in 3 mL of DMF, followed by the addition of 15 mL of MeCN. Next 0.1 mL of acetic acid was added, and then bpe (0.185 g, 0.014 mmol, a 3-fold excess) was dissolved in this solution. After one day black crystals formed, which were collected by filtration, washed with MeCN (2 portions by 3 mL) and dried on air. Yield 0.07 g (40%). Anal, calc. for C_42_H_52_N_6_O_15_Fe_2_Mn / found, %: C 48.2/48.1, H 5.00/5.15, N 8.02/7.98.

#### 3.2.8. Synthesis of [MnFe_2_O(OAc)_6_(bpe)(DMF)]*_n_*·3.5*n*DMF (11·3.5DMF)

[Fe_2_MnO(OAc)_6_(H_2_O)_3_] (0.1 g, 0.169 mmol) was dissolved in 3 mL of DMF, followed by the addition of 15 mL of MeCN. Next 0.1 mL of acetic acid was added, and then bpe (0.038 g, 0.211 mmol, a 25% excess) was dissolved in this solution. After one day black crystals formed, which were collected by filtration, washed with MeCN (two portions of 3 mL) and dried on air. Yield 0.02 g (10%). Anal, calc. for C_34.5_H_52.5_N_5.5_O_16.5_Fe_2_Mn/found, %: C 42.5/42.1, H 5.43/5.35, N 7.90/7.95.

### 3.3. X-ray Structure Determination

For X-ray structure determination single crystals of the compounds **1**–**8** were isolated from the mother liquors and mounted on a Bruker APEX II diffractometer equipped with a CCD camera and a graphite monochromated MoK_α_ radiation source (λ = 0.71073 Å) at the N. S. Kurnakov Institute of General and Inorganic Chemistry (Moscow, Russia). X-ray structure determination for compounds **9**–**11** was performed using a a Kappa-Nonius four circle diffractometer equipped with a CCD camera and a graphite monochromated MoK_α_ radiation source (λ = 0.71073 Å), located at the Centre de Diffractométrie (CDIFX), Université de Rennes 1 (Rennes, France).

Effective absorption correction was performed using SCALEPACK. Structures of the complexes were solved by the direct method using SHELXS-97 [85] or Sir-97 [86] software, and refined with a full matrix least squares method on *F*^2^ using SHELXL-97, SHELX-2014 or SHELX-2018 program [87]. H atoms were treated by a riding model. Solvent molecules, which could not be localized, were removed by SQUEEZE procedure for compounds **9**–**11 [88]**. The structure of **6** was solved taking into account crystal twinning (Flack parameter is 0.42(4)). Crystallographic data and structure refinement parameters for **1**–**11** are presented in Table 2, Table 3 and Table 4. Supplementary crystallographic data for the compounds synthesized are given in CCDC numbers 2055493-2055503 for **1**–**11**, respectively. These data can be obtained free of charge from The Cambridge Crystallographic Data Centre via www.ccdc.cam.ac.uk/data_request/cif.

### 3.4. DFT Calculations

The signs and the magnitudes of exchange coupling parameters *J* were independently estimated by DFT calculations similar to the previously reported by us in details [89], brief description of the methodology is provided in this section.

The calculations were performed via ORCA software [90]. TPSSh [91,92,93] exchange-correlation potential was employed for the calculation together with LANLTZ basis sets [94] for 3d ions and def2-SVP [95] basis set for the rest of the elements.

The atomic coordinates were taken from the crystallographic data. For calculation of each *J* value, all Mn^2+^ ions in Mn_4_ core except the two ions taking part in the coupling were “substituted” by diamagnetic Zn^2+^ ions in order to simplify the system of spin states of the species. Broken symmetry DFT approach was applied for the calculation of *J*: first, a single-point calculation was performed for the high-spin state of the Mn_2_Zn_2_ species, then the broken symmetry state was constructed by artificial flipping the spin projections of the unpaired electrons localized on one of the Mn^2+^ ions, and a single-point calculation was performed again. *J* value was obtained using the energies *E*_HS_ and *E*_BS_ resulted from the two converged single-point calculations as *J* = − (*E*_HS_ − *E*_BS_)/(<*S*^2^>_HS_−<*S*^2^>_BS_) [96] (<S^2^>_HS_ and <S^2^>_BS_—the total spin operator expectation values derived from the calculations).

## 4. Conclusions

In this study a variety of transformations of Mn-containing complexes in reactions with N-donor heterocycles was shown. In was found that homometallic Mn pivalates underwent metamorphosis while heterometallic acetates with a Fe_2_MnO core preserved their structure, giving rise to coordination polymers. Different behavior of Mn^II^ or Ni^II^ and Co^II^ pivalates in reactions with FeCl_3_ was also revealed: while the first complex produced a hexanuclear [Mn^II^_4_Fe^III^_2_O_2_(Piv)_10_(MeCN)_2_(HPiv)_2_] compound, the latter (Ni^II^ and Co^II^) pivalates under similar conditions gave trinuclear pivalates with a Fe_2_MnO core.

Several new coordination polymers with bridging pyrimidine (i.e., [Mn_6_O_2_(Piv)_10_(pym)_2_]*_n_*), 4,4-bipyridine (i.e., [MnFe_2_O(OAc)_6_(4,4′-bipy)_2_]*_n_*) or 1,2-*bis*-trans-(4-pyridyl)ethylene (i.e., [MnFe_2_O(OAc)_6_(bpe)_2_]*_n_* and [MnFe_2_O(OAc)_6_(bpe)(DMF)]*_n_*) were prepared. Unexpectedly, the composition of the coordination polymers based on MnFe_2_O(OAc)_6_ was probably governed by a fine balance between formation and crystallization kinetics and solubility of certain species, in contrast to the ratio of potential vacations in the coordination spheres of metal ions and the number of donor atoms). The crystal lattice of [MnFe_2_O(OAc)_6_(bpe)(DMF)]*_n_* collapsed upon desolvatation, and the resulting compound was not porous in respect to N_2_, however the porosity was restored upon interaction with methanol and ethanol. Notably, the hexanuclear unit {Mn^II^_4_Fe^III^_2_O_2_(O_2_CR)_10_} can be considered a promising new building block for the creation of coordination polymers.

Magnetic properties of the compounds were in line with those expected for Mn carboxylates. The temperature dependence of the magnetic susceptibility of [Mn_4_(μ_3_-OH)(Piv)_7_(μ-pz)_2_]*_n_* could be fitted with five exchange coupling parameters, and their reliability was independently checked by DFT calculations.

## Data Availability

The data presented in this study are available on request from the corresponding author.

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
