# Peer review of "Versatile Reactivity of MnII Complexes in Reactions with N-Donor Heterocycles: Metamorphosis of Labile Homometallic Pivalates vs. Assembling of Endurable Heterometallic Acetates"

_molecules, 2021, doi:10.3390/molecules26041021_

Round 1

Reviewer 1 Report

Eleven new manganese complexes, including manganese coordination polymers, dimers, tetramers and hexamers, were reported in this paper. The authors studied the experimental conditions (inert atmosphere and /or prolonged standing in air) needed to prepare this variety of manganese compounds. The prepared compounds were characterized by elemental analyses and IR spectroscopy, while their crystal structures were determined by single crystal X-ray structure analysis. The authors also studied magnetic properties of the prepared compounds, while the structure of one compound (11) was porous, so the authors studied the gas and liquid sorption properties of that compound.

I’ve carefully read this paper and found it interesting and worthy of publication in Molecules, but believe the paper is way too extensive for publication as a single paper. There is too much material (11 compounds, their structures, syntheses and properties) presented, so I was losing my focus from time to time during the reading (the readers might experience the same), regardless of the fact that the paper is well-written and the presentation of the results is nice. I would suggest the authors to make two papers: one dealing with the manganese dimers, tetramers and hexamers and their structures and magnetic properties; the other dealing with manganese coordination polymers and their structures, with special attention to gas and liquid sorption properties. Nevertheless, should the authors still want to publish all these results in a single paper, it is fine with me. However, some revisions are necessary prior to publication.

1) The authors discuss π-π interactions quite often, citing only the respective centroid-centroid distances. They should also provide the slippage (offset) values for all the discussed π-π interactions.

2) Usually, the authors are required to resolve and/or explain all the alerts A and B in the checkCIF reports. There are a lot of alerts A for the structures of 9, 10 and 11 and a lot of alerts B for the structures of 3, 6, 8, 9, 10 and 11. The authors should check these alerts and try to resolve them. If they don’t succeed in resolving, they should provide adequate explanations of the alerts and the efforts made. The explanations should be inserted in the respective CIF files by VRF instruction, so they also appear in the checkCIF reports, following the concerning alerts.

Author Response

COMMENT

Eleven new manganese complexes, including manganese coordination polymers, dimers, tetramers and hexamers, were reported in this paper. The authors studied the experimental conditions (inert atmosphere and /or prolonged standing in air) needed to prepare this variety of manganese compounds. The prepared compounds were characterized by elemental analyses and IR spectroscopy, while their crystal structures were determined by single crystal X-ray structure analysis. The authors also studied magnetic properties of the prepared compounds, while the structure of one compound (11) was porous, so the authors studied the gas and liquid sorption properties of that compound.

I’ve carefully read this paper and found it interesting and worthy of publication in Molecules, but believe the paper is way too extensive for publication as a single paper. There is too much material (11 compounds, their structures, syntheses and properties) presented, so I was losing my focus from time to time during the reading (the readers might experience the same), regardless of the fact that the paper is well-written and the presentation of the results is nice. I would suggest the authors to make two papers: one dealing with the manganese dimers, tetramers and hexamers and their structures and magnetic properties; the other dealing with manganese coordination polymers and their structures, with special attention to gas and liquid sorption properties. Nevertheless, should the authors still want to publish all these results in a single paper, it is fine with me. However, some revisions are necessary prior to publication.

REPLY

We thank the Reviewer for his/her comments. We prefer to publish all results as a single paper, because in our opinion such combination of the results gives better impression of versatile reactivity of Mn-containing complexes.

COMMENT

1) The authors discuss π-π interactions quite often, citing only the respective centroid-centroid distances. They should also provide the slippage (offset) values for all the discussed π-π interactions.

REPLY

The values of slippage were calculated for compounds 1, 2, 10. These values were added to the text of the revised manuscript.

COMMENT

2) Usually, the authors are required to resolve and/or explain all the alerts A and B in the checkCIF reports. There are a lot of alerts A for the structures of 9, 10 and 11 and a lot of alerts B for the structures of 3, 6, 8, 9, 10 and 11. The authors should check these alerts and try to resolve them. If they don’t succeed in resolving, they should provide adequate explanations of the alerts and the efforts made. The explanations should be inserted in the respective CIF files by VRF instruction, so they also appear in the checkCIF reports, following the concerning alerts.

REPLY

We carefully revised X-ray structures of 3, 6, 8, 9, 10 and 11 and we managed to decrease the number of A and B-level alerts. In particular, for compound 3 the number of B-level alerts was 1; after revision the number of B-level alerts is 0.

For compound 8 there were 2 B-level alerts, their origin could cause questions regarding the precision of X-ray structure refinement:

Disordered C29 has ADP max/min Ratio ..... 4.1 Note

Large Average Ueq of Residue Including N2S 0.170 Check

After revision the number of B-level alerts for compound 8 is equal to 2:

Low ’MainMol’ Ueq as Compared to Neighbors of C2 Check

Low ’MainMol’ Ueq as Compared to Neighbors of C17 Check

These alerts are associated with rotation of tert-butyl groups.

For 9 the number of A-level alerts was 5 and of B-level alerts was 9; after revision the number of A-level alerts is 1 and of B-level alerts is 3.

For 10 the number of A-level alerts was 3 and of B-level alerts was 15; after revision the number of A-level alerts is 0 and of B-level alerts is 2.

for 11 the number of A-level alerts was 3 and of B-level alerts was 6; after revision the number of A-level alerts is 1 and of B-level alerts is 4.

The situation was more complicated for compound 6. In this case the number of B-level alerts was 2, however one of the alerts suggested twin refinement:

The Flack x is >> 0 - Do a BASF/TWIN Refinement

We performed twin refinement and after revision the number of B-level alerts was 8. However, 7 of these alerts are caused by rotation of tert-butyl group, and the remaining one is caused by low crystal quality (this last alert regarding low precision of C-C bonds was the same in the previous version). Despite larger quantity of B-level alerts after refinement, we prefer to leave the revised version of X-ray structure refinement because it is more correct, in our opinion.

All respective changes were made in tables 1-3, the values of bond lengths and some angles in the text were also corrected. It was also mentioned that the structure 6 was solved taking into account crystal twinning (Flack parameter is 0.42(4)).

Reviewer 2 Report

This very interesting manuscript reports the synthesis, characterization and diverse structural features of a new and very large series of Mn(II) complexes and coordination polymers containing pivalate/acetate and phen or bipy blocks. The numerous results obtained in the present work (11 crystal structures) are clearly of very high scientific significance with a considerable potential to be explored in further research. Also, notable magnetic behavior and sorption properties were studied for some compounds. This manuscript extends the authors' well-known prior research on metal-carboxylate coordination compounds and is excellently organized and presented, technically correct and supported by detailed data. The subject of this study is of high importance and contributes to widening the family of polynuclear manganese complexes and coordination polymers.

By providing an attractive blend of coordination chemistry with crystal engineering and molecular magnetism, this very appealing work also well fits the scope of Molecules. I therefore highly recommend the publication of this manuscript.

A few minor points can be addressed in a revised version.

1. Please check if all abbreviations are explained.

2. It would be good to add in the introduction a scheme containing the formulae of all organic ligands present in the obtained compounds (piv, bipy, phen, pz, dpe, etc)

3. TGA. Fig. 13. Please indicate a heating rate.

4. Table 1. A line for c parameter is in bold. Please check the formatting.

5. In Fig. 1, some formulae of coordination compounds are not in [] brackets but in the text the brackets are used. Please unify.

6. In the beginning of introduction, when talking about diverse applications of polynuclear coordination compounds, some additional studies on polynuclear metal-complex catalysts [Catalysis Reviews, 2012, 54, 1-40; Chem. Eur. J. 2015, 21, 8758-8770; Advances Inorg. Chem. 2013, 65, 1-31] and magnetic materials [Coord. Chem. Rev. 2019, 398, 213015; Coord. Chem. Rev. 2017, 339, 17-103] are suggested to be referenced.

7. Graphical abstract is not provided (not available to this reviewer).

Author Response

COMMENT

This very interesting manuscript reports the synthesis, characterization and diverse structural features of a new and very large series of Mn(II) complexes and coordination polymers containing pivalate/acetate and phen or bipy blocks. The numerous results obtained in the present work (11 crystal structures) are clearly of very high scientific significance with a considerable potential to be explored in further research. Also, notable magnetic behavior and sorption properties were studied for some compounds. This manuscript extends the authors' well-known prior research on metal-carboxylate coordination compounds and is excellently organized and presented, technically correct and supported by detailed data. The subject of this study is of high importance and contributes to widening the family of polynuclear manganese complexes and coordination polymers.

By providing an attractive blend of coordination chemistry with crystal engineering and molecular magnetism, this very appealing work also well fits the scope of Molecules. I therefore highly recommend the publication of this manuscript.

A few minor points can be addressed in a revised version.

  1. Please check if all abbreviations are explained.

REPLY

We thank the Reviewer for his/her comments.

We added explanation of abbreviation pym in the abstract. All other abbreviations were explained.

COMMENT

  1. It would be good to add in the introduction a scheme containing the formulae of all organic ligands present in the obtained compounds (piv, bipy, phen, pz, dpe, etc)

REPLY

We added a Scheme with formulae of all mentioned organic ligands.

COMMENT

  1. TGA. Fig. 13. Please indicate a heating rate.

REPLY

Heating rate was 5°C per minute. This indication was added to Experimental section.

COMMENT

  1. Table 1. A line for c parameter is in bold. Please check the formatting.

REPLY

Formatting was corrected.

COMMENT

  1. In Fig. 1, some formulae of coordination compounds are not in [] brackets but in the text the brackets are used. Please unify.

REPLY

Brackets were used in the text in order to separate solvate molecules. Since solvates were omitted on Fig. 1, some of the brackets were not shown.

We agree that such discrepancy may be confusing for a reader. We added brackets to Fig. 1.

COMMENT

  1. In the beginning of introduction, when talking about diverse applications of polynuclear coordination compounds, some additional studies on polynuclear metal-complex catalysts [Catalysis Reviews, 2012, 54, 1-40; Chem. Eur. J. 2015, 21, 8758-8770; Advances Inorg. Chem. 2013, 65, 1-31] and magnetic materials [Coord. Chem. Rev. 2019, 398, 213015; Coord. Chem. Rev. 2017, 339, 17-103] are suggested to be referenced.

REPLY

These references were added (refs. 3-5 and 12-13 in the revised version).

COMMENT

  1. Graphical abstract is not provided (not available to this reviewer).

REPLY

Graphical abstract is now provided along with the revised version of the manuscript.